# FEDDUET: BRIDGING MODALITY GAPS WITH DECOUPLED UNCERTAINTY-ENHANCED TRAINING

## ABSTRACT

Federated learning enables collaborative model training for multimodal health sensing while preserving data privacy. A critical challenge, however, is modality heterogeneity, which manifests along two axes: *intra-client instability*, caused by per-sample sensor dropouts, and *inter-client heterogeneity*, driven by differences in clients' sensor suites. Existing federated methods often rely on oversimplified assumptions about missing data and fail to capture these complex dynamics. We address this gap by introducing a realistic problem formulation and a principled simulation framework. Building on this foundation, we propose FedDUET (Decoupled Uncertainty-Enhanced Training), an approach designed to handle both axes of modality heterogeneity. To mitigate intra-client instability, FedDUET leverages an Uncertainty-as-Temperature (UT) loss to dynamically calibrate predictions based on data uncertainty. To manage inter-client heterogeneity, it employs a Decoupled Training (DT) strategy that specializes a private model head for each client's unique sensor suite while isolating the shared representation to preserve its generalizability. Across four real-world multimodal sensing datasets and diverse heterogeneity regimes, FedDUET achieves state-of-the-art performance. Our results highlight that explicitly modeling uncertainty and decoupling generalization from personalization are essential principles for making multimodal federated learning robust in real-world settings.

## 1 INTRODUCTION

Healthcare sensing increasingly relies on multimodal time-series data from wearable and embedded devices (Ramachandram & Taylor, 2017; Narayanswamy et al., 2024) to enable applications such as activity recognition (Reiss & Stricker, 2012), eating detection (Shin et al., 2022), emotion inference (Park et al., 2020), and stress monitoring (Schmidt et al., 2018). Federated Learning (FL) (McMahan et al., 2017; Kairouz et al., 2021) is a natural fit for this domain, allowing models to train on sensitive user data without it ever leaving the device. Yet, this vision is undermined by a fundamental real-world challenge: pervasive modality heterogeneity (Feng et al., 2023). This problem degrades model performance along two distinct axis *(i) intra-client instability*, where an individual's sensors experience dynamic, intermittent dropouts from issues like battery drain or connectivity loss (Xu et al., 2025); and *(ii) inter-client heterogeneity*, where the set of available sensors is static but varies across users with different devices (Ouyang et al., 2023).

Despite its prevalence, this dual-axis modality heterogeneity problem remains largely unaddressed. Prior FL methods rely on oversimplified models, either neglecting the temporal, bursty nature of sensor dropouts (Feng et al., 2023) or assuming purely static differences between clients (Zhao et al., 2022; Bao et al., 2023). This critical gap impedes the development of truly robust algorithms. Our first contribution is to formalize this challenge and introduce a principled framework for simulating modality heterogeneity. The framework models intra-client instability with a two-state Markov chain to generate bursty, temporal dropouts, and inter-client heterogeneity with a Beta-Bernoulli process to simulate diverse client populations, as demonstrated in Figure 1.

Within this challenging paradigm, we propose **D**ecoupled **U**ncertainty-**E**nhanced **T**raining, FedDUET, a method to tackle modality heterogeneity with two synergistic components. First, to combat intra-client instability, FedDUET employs *Uncertainty-as-Temperature (UT)* loss. This mechanism estimates the aleatoric uncertainty of each input and uses it as a temperature to scale the model's

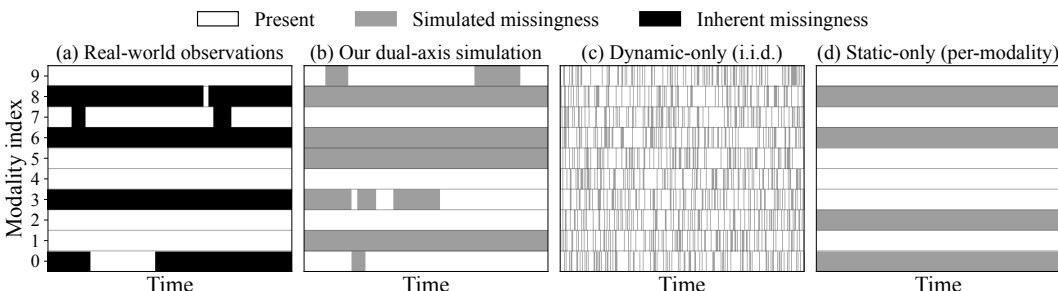

Figure 1: Comparison of missingness patterns. (a) Real-world multimodal health sensing data from the Opportunity dataset (Roggen et al., 2010) exhibits a mixture of static unavailability (inter-client heterogeneity) and dynamic, bursty dropouts (intra-client instability). (b) Our simulation framework faithfully reproduces these complex dual-axis patterns. In contrast, conventional models rely on simplified assumptions, capturing only (c) i.i.d. dynamic dropouts (Feng et al., 2023) or (d) purely static client differences (Bao et al., 2023).

logits. Specifically, UT modulates the model's predictive entropy, steering the model prediction toward a distribution that better reflects the true posterior under intra-client instability. Second, to tackle inter-client heterogeneity, FedDUET adopts *Decoupled Training (DT)* strategy. This approach features a hybrid architecture with shared, general-purpose components and a private, specialized head for each client. Crucially, the training leverages this split: the shared model learns to produce generalizable feature representations and reliable uncertainty estimates, and these estimates directly temper the private head's training objective. By decoupling these private updates, the process allows the head to specialize effectively without corrupting the shared model's generalizable knowledge. We provide a comprehensive discussion of related work and situate our contributions within the broader literature in Appendix B.

We empirically evaluate FedDUET against six baselines across three real-world multimodal health sensing datasets, employing our simulation framework to generate realistic modality-heterogeneity patterns. Across diverse heterogeneity regimes, FedDUET consistently outperforms baselines, achieving absolute macro-F1 score improvements of 1.52%~6.49%. We further validate its effectiveness on a dataset with inherent missingness, where it also achieves the best performance.

Our contributions are as follows:

- **Dual-axis modality heterogeneity simulation framework**. We provide a realistic formalization and principled simulation framework for the dual-axis modality heterogeneity problem in multimodal health sensing FL, capturing both intra-client instability and inter-client heterogeneity.
- **The FedDUET method**. We propose FedDUET, a novel method that integrates an Uncertainty-as-Temperature loss to enhance robustness to intra-client instability and a Decoupled Training strategy to enable adaptation under inter-client heterogeneity.
- **Empirical validation**. We conduct extensive empirical evaluations showing that FedDUET achieves state-of-the-art performance, with absolute macro-F1 improvements ranging from 1.52% to 6.49% over the baselines across diverse heterogeneity regimes.

## 2 A PRINCIPLED FRAMEWORK FOR SIMULATING MODALITY HETEROGENEITY

Existing federated learning methods for multimodal sensing (Zhao et al., 2022; Bao et al., 2023; Feng et al., 2023) are constrained by unrealistic missingness simulations. We address this gap by introducing a principled simulation framework that formalizes the two orthogonal axes of real-world modality heterogeneity: *(i) intra-client instability* and *(ii) inter-client heterogeneity*.

The fidelity of our simulation framework is illustrated in Figure 1. Real-world multimodal health sensing data (a) exhibit both permanently absent modalities and others that drop out dynamically in temporally correlated, bursty segments (Roggen et al., 2010). Our simulation (b) reproduces these complex patterns, in contrast to naïve approaches that assume (c) simplistic i.i.d. dynamic

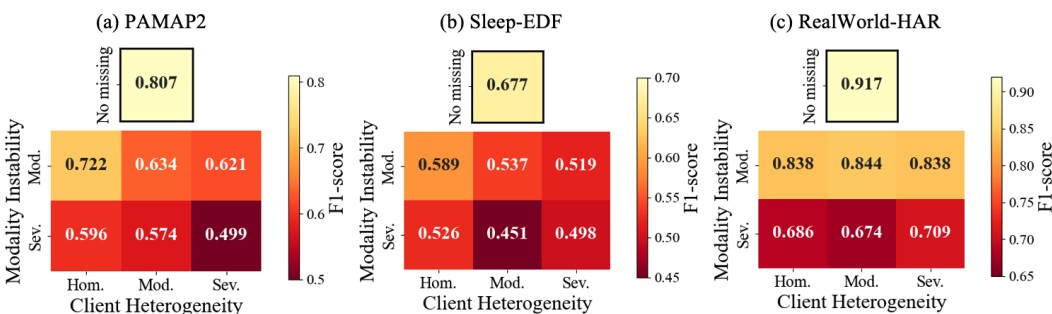

Figure 2: F1-scores on PAMAP2, Sleep-EDF, and RealWorld-HAR datasets under the federated learning setting with FedAvg (McMahan et al., 2017) algorithm. The top shows performance with complete data (no missing), while the heatmaps depict the degradation under increasing intra-client modality instability level {Moderate, Severe} and inter-client modality heterogeneity level {Homogeneous, Moderate, Severe}.

dropouts (Feng et al., 2023) or (d) purely static modality availability (Bao et al., 2023; Zhao et al., 2022). This realistic behavior arises from jointly modeling the two orthogonal axes of modality heterogeneity, as detailed below.

## 2.1 MODELING INTRA-CLIENT INSTABILITY

To capture the bursty, temporal nature of modality instability within a client, we model the operational status of each sensor with a two-state Markov chain. This approach effectively simulates periods of sustained sensor availability or failure, because the Markovian property gives each state persistence, discouraging random changes at each timestep. More formally, for each present modality $m$ on client $k$, we define a binary state $s_{t,k,m}$ indicating if the sensor is operational at time $t$:

$$s_{t,k,m} = \begin{cases} 1, & \text{if modality } m \text{ is operational at time } t, \\ 0, & \text{otherwise.} \end{cases}$$

Transitions between states are governed by a matrix $\mathbf{P}$:

$$\mathbf{P} = \begin{bmatrix} p_{00} & p_{01} \\ p_{10} & p_{11} \end{bmatrix},$$

where $p_{ij}$ is the probability of transitioning from state $i$ to state $j$ (0: missing, 1: present). By tuning these dataset-specific probabilities, we can simulate varying levels of instability, from moderate (intermittent) to severe (long, bursty) sensor dropouts.

## 2.2 MODELING INTER-CLIENT HETEROGENEITY

To capture client heterogeneity—the static differences in sensor suites across a population, we employ a Beta–Bernoulli process. This principled approach models the real-world scenario where each user's device ownership is drawn from a broader population distribution. First, to model latent client-level sensor availability, we sample a probability $p_{a,k}$ from a Beta distribution:

$$p_{a,k} \sim \text{Beta}(\alpha_a, \beta_a),$$

where the hyperparameters $(\alpha_a, \beta_a)$ control the level of heterogeneity in the environment. A severe heterogeneity setting is created by centering the Beta distribution's mean at 0.5 (by setting $\alpha_a \approx \beta_a$), which maximizes the combinatorial diversity of sensor suites across clients. Conversely, a moderate heterogeneity setting is achieved by shifting the distribution's mean away from 0.5 (by using unbalanced $\alpha_a$ and $\beta_a$ values). This creates a more uniform population where clients have a more consistent set of available sensors, thereby reducing the overall variation in their device configurations.

Next, the specific sensor suite for client $k$ is determined by sampling a binary indicator $\delta_{m,k}$ for each modality $m$ from a Bernoulli distribution parameterized by the client's unique $p_{a,k}$:

$$\delta_{m,k} \sim \text{Bernoulli}(p_{a,k}), \quad \delta_{m,k} = \begin{cases} 1, & m \text{ is available,} \\ 0, & \text{otherwise.} \end{cases}$$

The final set of available modalities for client $k$ is thus $\mathcal{M}_{\text{available},k} = \{m \mid \delta_{m,k} = 1\}$.

**Integrated Simulation Process.** Our integrated simulation process first establishes each client's static hardware profile via the inter-client heterogeneity model and then simulates dynamic sensor failures using the intra-client instability model. As illustrated in Figure 2, applying our simulation framework quantifies the impact of realistic modality heterogeneity on model performance. The systematic performance degradation (e.g., a drop exceeding 30% on PAMAP2 under severe conditions) underscores the importance of accounting for these real-world conditions. This demonstrates that our framework can generate challenging scenarios for standard algorithms such as FedAvg (McMahan et al., 2017), thereby serving as a valuable testbed for developing and evaluating more robust methods. Detailed hyperparameter configurations and additional examples are provided in Appendix D.

## 3 PRELIMINARIES: FEDERATED LEARNING

Federated Learning (FL) is a distributed machine learning paradigm where a central server coordinates a set of $K$ clients to train a shared global model (McMahan et al., 2017; Kairouz et al., 2021). Each client $k \in \{1, \dots, K\}$ holds a private dataset $\mathcal{D}_k$ that is never shared, preserving data locality and privacy. The objective is to learn a single set of global model parameters $\theta$ that minimizes a weighted sum of the local loss functions across all clients:

$$\min_{\theta} \ \mathcal{L}(\theta) \ = \ \sum_{k=1}^{K} w_k \, \mathcal{L}_k(\theta), \tag{1}$$

where $\mathcal{L}_k(\theta)$ is the loss on client $k$'s data $\mathcal{D}_k$, and $w_k$ is the weight assigned to client $k$.

The fundamental algorithm for this task is Federated Averaging (FedAvg) (McMahan et al., 2017). It proceeds in synchronous communication rounds $t = 0, 1, \dots$. In each round, the server broadcasts the current global parameters $\theta^t$ to a subset of clients $\mathcal{S}^t$. Each selected client $k \in \mathcal{S}^t$ performs local optimization to produce updated parameters $\theta_k^{t+1}$. The server then aggregates these returned models by computing a weighted average to obtain the next global model:

$$\theta^{t+1} \ = \ \sum_{k \in \mathcal{S}^t} \tilde{w}_k \, \theta_k^{t+1}, \qquad \tilde{w}_k \ = \ \frac{|\mathcal{D}_k|}{\sum_{j \in \mathcal{S}^t} |\mathcal{D}_j|}. \tag{2}$$

While FedAvg provides a general-purpose solution, it does not explicitly account for the challenges of modality heterogeneity in multimodal health sensing.

## 4 THE FEDDUET METHOD

We now introduce `FedDUET`, an approach designed to tackle modality heterogeneity through the integration of two synergistic components. At the sample level, the *Uncertainty-as-Temperature (UT) loss* (Section 4.1) provides a fine-grained mechanism to handle the uncertainty arising from intra-client sensor dropouts. This mechanism is embedded within a *Decoupled Training (DT) strategy* (Section 4.2), which manages inter-client heterogeneity.

The client-side training process is illustrated in Figure 3. Before detailing the loss functions, we first introduce the core components of the architecture:

- **Encoders and Fusion**: Each input modality $(x_1, x_2, \dots)$ is processed by a dedicated `Encoder` to produce a unimodal feature representation $(h_1, h_2, \dots)$. These features are then fused by a `Fusion` module to form a unified multimodal representation $h_f$.
- **Uncertainty Heads**: Running in parallel, lightweight `Uncertainty Heads` also process the unimodal features $(h_m)$. Their role is to estimate the uncertainty of each modality's data, outputting a scalar uncertainty estimate $(\sigma_m)$ and logits for the unimodal prediction task $(z_m)$.

Figure 3: The `FedDUET` client-side training process. Unimodal inputs ($x_m$) are processed by shared `Encoders` to produce features ($h_m$). These features are used in two parallel streams: (1) `Uncertainty Heads`, trained with unimodal losses ($\mathcal{L}_{\text{UT},m}$), estimate data uncertainty and produce uncertainty scores ($\sigma_m$), and (2) a `Fusion` module creates a multimodal representation ($h_f$). The shared `G-Head` learns a general model, while the private `P-Head` specializes for the client. Crucially, the `P-Head`'s training objective ($\mathcal{L}_{\text{mUT}}$) is tempered by the fused uncertainty ($\sigma_f$), and stop-gradient (`sg`) operation detaches gradients for effective decoupled training.

- **Shared and Private Heads**: The model has two multimodal prediction heads. The `G-Head` (Global) is a shared component that learns a generalizable prediction from the fused representation $h_f$. The `P-Head` (Private) is a client-specific component that learns a personalized prediction, also from the fused representation.
- **Stop-Gradients**: The `sg` markers indicate where we apply stop-gradients to enable decoupled training, which is explained in Section 4.2.

This architecture forms the foundation for our specialized training objectives, which we detail next.

## 4.1 Uncertainty as Temperature for Intra-Client Modality Instability

The primary challenge of intra-client instability is that intermittent sensor dropouts introduce unreliable samples into the training data. This naturally raises the question of how such missing inputs affect the model's predictive distribution. Our intuition is that *the presence of missing data increases the entropy of the predictive posterior: as information decreases, the predictive distribution should flatten toward uniformity.* Appendix C formally proves this intuition, showing that the posterior entropy under missing inputs is higher than under complete observations.

Building on this result, we introduce the *Uncertainty-as-Temperature (UT) loss*. This mechanism implements this principle by scaling the model's logits with a learned, per-sample temperature ($\sigma$) derived from the input's estimated aleatoric uncertainty. This allows the model to dynamically modulate its own confidence: for uncertain inputs, it learns to increase $\sigma$ to soften the predictive distribution, while for high-quality inputs, it decreases $\sigma$ to sharpen its confidence.

This principle of learning input-dependent variance to mitigate data noise shares foundations with recent work in other domains. While prior work leveraged per-sample uncertainty, their objective has typically been to down-weight uncertain samples and reduce their influence on model updates (Kendall & Gal, 2017; Collier et al., 2021; Englesson et al., 2023). In contrast, our approach uses uncertainty to modulate predictive entropy, steering the model toward a distribution that better reflects the true posterior. Uncertainty-as-Temperature loss thereby provides robustness against *intra-client instability* by aligning predictive confidence with data uncertainty.

In particular, as illustrated in Figure 3, a dedicated `Uncertainty Head` predicts the log-variance $s_m = \log \sigma_m^2$ of sample $x_m$. The resulting standard deviation $\sigma_m = \exp(s_m/2)$ is then used to temper the logits, defining the unimodal UT loss:

$$\mathcal{L}_{\text{UT},m} = CE\left(\frac{z_m}{\sigma_m}, y\right). \tag{3}$$

These unimodal uncertainties are then fused into a multimodal uncertainty, $\sigma_f$, using a Bayesian precision-weighted scheme (Gelman et al., 1995):

$$\sigma_f = \left(\sum_{m=1}^{M} \frac{a_m}{\sigma_m^2 + \epsilon}\right)^{-1/2}, \tag{4}$$

where $a_m \in 0, 1$ denotes the availability of modality $m$ and $\epsilon$ is a stability constant.

This fused uncertainty is used in the multimodal UT loss:

$$\mathcal{L}_{\text{mUT}} = CE\left(\frac{z}{\sigma_f}, y\right), \tag{5}$$

which plays a critical, synergistic role in guiding the personalized component of our decoupled training, as described next.

## 4.2 A Decoupled Training for Inter-Client Modality Heterogeneity

While the UT loss addresses sample-level instability, a separate mechanism is needed to handle *inter-client heterogeneity*, where clients possess different static sets of sensors. A monolithic, end-to-end model is suboptimal for this challenge, as it forces the shared parameters to learn conflicting representations (Li et al., 2020) from clients with disparate data modalities. Our intuition is to resolve this conflict by structurally separating the model into shared components that capture general knowledge and a private component that specializes for each client's unique sensor suite.

To realize this, we employ a *Decoupled Training (DT)* strategy. As illustrated in Figure 3, this approach adopts a hybrid architecture with two distinct sets of parameters.

- Shared Components ($\theta_G$): A set of unimodal `Encoders`, their corresponding `Uncertainty Heads`, a multimodal `Fusion` module, and a global `G-Head`. These components are shared across all clients to learn a generalized representation and to serve as a reliable estimator of uncertainty.
- Private Component ($\theta_{P,k}$): A client-specific `P-Head` that is not shared and adapts to the client's local data and unique modality combinations.

The core of the DT mechanism is the isolation of these components during training. To prevent client-specific updates from corrupting the shared model, gradients from the private objective are detached from the shared components via a stop-gradient operation. Note that the architectural principle of decoupling a model into shared and private parts is well established in personalized federated learning. Foundational works like FedPer (Arivazhagan et al., 2019) separate a model into shared base and private personalization layers. More advanced methods such as FedRoD (Chen & Chao, 2022) also use a dual-head design to bridge generic and personalized learning, though their focus is on unimodal data and non-IID class distributions.

While these methods established the benefits of decoupling, our novelty lies in leveraging this separation to specifically address modality heterogeneity through *synergistic, uncertainty-guided personalization*. Unlike prior work where the private head learns only from the feature representation, our private `P-Head` is explicitly guided by the fused uncertainty estimate ($\sigma_f$, in Equation 4) provided by the shared model. This creates a powerful synergy: the shared model assesses input reliability, while the private head adapts not only to the client's available modalities but also to their real-time instability.

This synergy is formalized in our training objectives. The shared components are optimized with a composite loss, $\mathcal{L}_G$, to learn accurate and well-calibrated representations:

$$\mathcal{L}_{\text{G}} = \text{CE}(\boldsymbol{z}_G, y) + \frac{1}{M} \sum_{m=1}^{M} \mathcal{L}_{\text{UT},m}, \tag{6}$$

where $\boldsymbol{z}_G$ are the logits from the shared `G-Head`, $y$ is the ground-truth label, $M$ is the number of modalities, and $\mathcal{L}_{\text{UT},m}$ is the unimodal UT loss from Equation 3. Concurrently, each private head `P-Head` ($\theta_{P,k}$) is trained using the multimodal UT loss, which is directly tempered by the shared model's uncertainty estimate, $\sigma_f$:

$$\mathcal{L}_{P,k} = \mathcal{L}_{\text{mUT}}(z_{P,k}, \sigma_f, y) \tag{7}$$

where the logits $z_{P,k}$ are produced by the private `P-Head` for client $k$ from the detached shared representation $h_f$, i.e., $z_{P,k} = \text{P-Head}_k(\text{detach}(h_f))$. This strategy enables each `P-Head` to specialize as an expert on its client's data, while being guided by the uncertainty-aware signals of the shared model. In doing so, it effectively addresses inter-client modality heterogeneity without corrupting the generalizable knowledge learned by the shared components. The full `FedDUET` algorithm is provided in Appendix 1.

Table 1: F1-score comparisons with baselines under varying modality heterogeneity settings. We evaluate performance across inter-client heterogeneity (**H**) levels {Homogeneous, Moderate, Severe} and intra-client instability (**I**) levels {Moderate, Severe}. Results are averaged over five random seeds on three datasets, with the best results marked in **bold**.

| Method | H = Homogeneous | | | H = Moderate | | | H = Severe | | | Average |
|---|---|---|---|---|---|---|---|---|---|---|
| | I=Mod. | I=Sev. | Avg. | I=Mod. | I=Sev. | Avg. | I=Mod. | I=Sev. | Avg. | |
| FedAvg | $0.722_{\pm 0.006}$ | $0.596_{\pm 0.023}$ | $0.659_{\pm 0.014}$ | $0.634_{\pm 0.007}$ | $0.574_{\pm 0.007}$ | $0.604_{\pm 0.007}$ | $0.621_{\pm 0.007}$ | $0.499_{\pm 0.012}$ | $0.560_{\pm 0.009}$ | $0.608_{\pm 0.010}$ |
| FedProx | $0.722_{\pm 0.003}$ | $0.600_{\pm 0.019}$ | $0.661_{\pm 0.011}$ | $0.636_{\pm 0.003}$ | $0.569_{\pm 0.011}$ | $0.602_{\pm 0.007}$ | $0.620_{\pm 0.010}$ | $0.497_{\pm 0.014}$ | $0.559_{\pm 0.012}$ | $0.607_{\pm 0.010}$ |
| MOON | $0.723_{\pm 0.010}$ | $0.610_{\pm 0.010}$ | $0.687_{\pm 0.010}$ | $0.636_{\pm 0.013}$ | $0.561_{\pm 0.016}$ | $0.599_{\pm 0.014}$ | $0.616_{\pm 0.007}$ | $0.494_{\pm 0.012}$ | $0.555_{\pm 0.009}$ | $0.607_{\pm 0.011}$ |
| FedPer | $0.694_{\pm 0.010}$ | $0.453_{\pm 0.010}$ | $0.574_{\pm 0.010}$ | $0.644_{\pm 0.011}$ | $0.490_{\pm 0.005}$ | $0.567_{\pm 0.008}$ | $0.614_{\pm 0.010}$ | $0.440_{\pm 0.008}$ | $0.527_{\pm 0.009}$ | $0.556_{\pm 0.009}$ |
| Fed-RoD | $0.754_{\pm 0.003}$ | $0.609_{\pm 0.011}$ | $0.682_{\pm 0.007}$ | $0.656_{\pm 0.011}$ | $0.587_{\pm 0.006}$ | $0.622_{\pm 0.009}$ | $0.642_{\pm 0.013}$ | $0.493_{\pm 0.011}$ | $0.568_{\pm 0.012}$ | $0.624_{\pm 0.009}$ |
| PmcmFL | $0.723_{\pm 0.007}$ | $0.605_{\pm 0.007}$ | $0.664_{\pm 0.007}$ | $0.643_{\pm 0.013}$ | $0.578_{\pm 0.018}$ | $0.611_{\pm 0.016}$ | $0.618_{\pm 0.005}$ | $0.514_{\pm 0.017}$ | $0.566_{\pm 0.011}$ | $0.614_{\pm 0.011}$ |
| **FedDUET** | $\mathbf{0.761_{\pm 0.005}}$ | $\mathbf{0.641_{\pm 0.009}}$ | $\mathbf{0.701_{\pm 0.007}}$ | $\mathbf{0.683_{\pm 0.011}}$ | $\mathbf{0.596_{\pm 0.022}}$ | $\mathbf{0.639_{\pm 0.016}}$ | $\mathbf{0.655_{\pm 0.005}}$ | $\mathbf{0.520_{\pm 0.013}}$ | $\mathbf{0.587_{\pm 0.009}}$ | $\mathbf{0.642_{\pm 0.011}}$ |

(a) PAMAP2.

| Method | H = Homogeneous | | | H = Moderate | | | H = Severe | | | Average |
|---|---|---|---|---|---|---|---|---|---|---|
| | I=Mod. | I=Sev. | Avg. | I=Mod. | I=Sev. | Avg. | I=Mod. | I=Sev. | Avg. | |
| FedAvg | $0.589_{\pm 0.009}$ | $0.526_{\pm 0.007}$ | $0.558_{\pm 0.008}$ | $0.537_{\pm 0.007}$ | $0.451_{\pm 0.012}$ | $0.494_{\pm 0.009}$ | $0.519_{\pm 0.007}$ | $0.498_{\pm 0.012}$ | $0.508_{\pm 0.010}$ | $0.520_{\pm 0.009}$ |
| FedProx | $0.589_{\pm 0.006}$ | $0.534_{\pm 0.006}$ | $0.562_{\pm 0.006}$ | $0.531_{\pm 0.007}$ | $0.438_{\pm 0.008}$ | $0.485_{\pm 0.007}$ | $0.512_{\pm 0.007}$ | $0.497_{\pm 0.009}$ | $0.504_{\pm 0.008}$ | $0.517_{\pm 0.007}$ |
| MOON | $0.594_{\pm 0.007}$ | $0.527_{\pm 0.006}$ | $0.561_{\pm 0.007}$ | $0.537_{\pm 0.003}$ | $0.448_{\pm 0.009}$ | $0.492_{\pm 0.006}$ | $0.514_{\pm 0.008}$ | $0.495_{\pm 0.011}$ | $0.505_{\pm 0.009}$ | $0.519_{\pm 0.007}$ |
| FedPer | $0.580_{\pm 0.010}$ | $0.510_{\pm 0.004}$ | $0.545_{\pm 0.007}$ | $0.520_{\pm 0.006}$ | $0.434_{\pm 0.008}$ | $0.477_{\pm 0.007}$ | $0.518_{\pm 0.004}$ | $0.463_{\pm 0.007}$ | $0.491_{\pm 0.006}$ | $0.504_{\pm 0.006}$ |
| Fed-RoD | $0.602_{\pm 0.005}$ | $0.531_{\pm 0.008}$ | $0.566_{\pm 0.007}$ | $0.532_{\pm 0.006}$ | $0.447_{\pm 0.010}$ | $0.489_{\pm 0.008}$ | $0.531_{\pm 0.014}$ | $0.502_{\pm 0.007}$ | $0.517_{\pm 0.010}$ | $0.524_{\pm 0.008}$ |
| PmcmFL | $0.601_{\pm 0.006}$ | $0.547_{\pm 0.005}$ | $0.574_{\pm 0.005}$ | $0.528_{\pm 0.004}$ | $0.439_{\pm 0.009}$ | $0.484_{\pm 0.006}$ | $0.489_{\pm 0.004}$ | $0.483_{\pm 0.004}$ | $0.486_{\pm 0.004}$ | $0.515_{\pm 0.005}$ |
| **FedDUET** | $\mathbf{0.616_{\pm 0.007}}$ | $\mathbf{0.557_{\pm 0.006}}$ | $\mathbf{0.586_{\pm 0.007}}$ | $\mathbf{0.552_{\pm 0.008}}$ | $\mathbf{0.467_{\pm 0.006}}$ | $\mathbf{0.509_{\pm 0.007}}$ | $\mathbf{0.540_{\pm 0.007}}$ | $\mathbf{0.518_{\pm 0.003}}$ | $\mathbf{0.529_{\pm 0.005}}$ | $\mathbf{0.542_{\pm 0.006}}$ |

(b) Sleep-EDF.

| Method | H = Homogeneous | | | H = Moderate | | | H = Severe | | | Average |
|---|---|---|---|---|---|---|---|---|---|---|
| | I=Mod. | I=Sev. | Avg. | I=Mod. | I=Sev. | Avg. | I=Mod. | I=Sev. | Avg. | |
| FedAvg | $0.838_{\pm 0.002}$ | $0.686_{\pm 0.007}$ | $0.762_{\pm 0.004}$ | $0.844_{\pm 0.005}$ | $0.674_{\pm 0.007}$ | $0.759_{\pm 0.006}$ | $0.838_{\pm 0.007}$ | $0.709_{\pm 0.007}$ | $0.773_{\pm 0.007}$ | $0.765_{\pm 0.006}$ |
| FedProx | $0.832_{\pm 0.009}$ | $0.673_{\pm 0.004}$ | $0.753_{\pm 0.006}$ | $0.847_{\pm 0.001}$ | $0.673_{\pm 0.010}$ | $0.760_{\pm 0.005}$ | $0.840_{\pm 0.003}$ | $0.716_{\pm 0.009}$ | $0.778_{\pm 0.006}$ | $0.763_{\pm 0.006}$ |
| MOON | $0.836_{\pm 0.013}$ | $0.670_{\pm 0.004}$ | $0.753_{\pm 0.008}$ | $0.844_{\pm 0.004}$ | $0.669_{\pm 0.007}$ | $0.757_{\pm 0.005}$ | $0.843_{\pm 0.005}$ | $0.709_{\pm 0.008}$ | $0.776_{\pm 0.007}$ | $0.762_{\pm 0.007}$ |
| FedPer | $0.852_{\pm 0.014}$ | $0.528_{\pm 0.014}$ | $0.690_{\pm 0.014}$ | $0.820_{\pm 0.014}$ | $0.616_{\pm 0.010}$ | $0.718_{\pm 0.012}$ | $0.835_{\pm 0.002}$ | $0.720_{\pm 0.010}$ | $0.778_{\pm 0.006}$ | $0.729_{\pm 0.011}$ |
| Fed-RoD | $\mathbf{0.883_{\pm 0.005}}$ | $0.699_{\pm 0.014}$ | $0.791_{\pm 0.010}$ | $0.856_{\pm 0.004}$ | $0.702_{\pm 0.010}$ | $0.779_{\pm 0.007}$ | $0.850_{\pm 0.008}$ | $0.751_{\pm 0.011}$ | $0.800_{\pm 0.009}$ | $0.790_{\pm 0.009}$ |
| PmcmFL | $0.840_{\pm 0.006}$ | $0.702_{\pm 0.011}$ | $0.771_{\pm 0.008}$ | $0.842_{\pm 0.005}$ | $0.666_{\pm 0.004}$ | $0.754_{\pm 0.005}$ | $0.834_{\pm 0.005}$ | $0.722_{\pm 0.012}$ | $0.778_{\pm 0.008}$ | $0.768_{\pm 0.007}$ |
| **FedDUET** | $0.867_{\pm 0.009}$ | $\mathbf{0.740_{\pm 0.004}}$ | $\mathbf{0.803_{\pm 0.007}}$ | $\mathbf{0.858_{\pm 0.007}}$ | $\mathbf{0.709_{\pm 0.005}}$ | $\mathbf{0.784_{\pm 0.006}}$ | $\mathbf{0.855_{\pm 0.007}}$ | $\mathbf{0.766_{\pm 0.003}}$ | $\mathbf{0.811_{\pm 0.005}}$ | $\mathbf{0.799_{\pm 0.006}}$ |

(c) RealWorld-HAR.

## 5 EXPERIMENTS

### 5.1 SETUP

**Datasets and baselines.** We use three publicly available multimodal health sensing datasets in our experiments: PAMAP2 (Reiss & Stricker, 2012), Sleep-EDF (Goldberger et al., 2000; Kemp et al., 2000) and RealWorld-HAR (Sztyler & Stuckenschmidt, 2016). We benchmark FedDUET against the foundational FedAvg (McMahan et al., 2017); methods for statistical non-IID data (FedProx (Li et al., 2020), MOON (Li et al., 2021)) to show modality heterogeneity is a distinct challenge; architecturally similar personalization methods (FedPer (Arivazhagan et al., 2019), Fed-RoD (Chen & Chao, 2022)); and PmcmFL (Bao et al., 2023), a direct competitor for modality-heterogeneous FL.

**Models and learning.** All methods share a common backbone: 1D CNNs (Haresamudram et al., 2022) serve as unimodal encoders, and a masked multi-context attention mechanism (Bahdanau et al., 2014) fuses available modality representations for classification by a two hidden-layer MLP. FedRoD (Chen & Chao, 2022) and FedDUET augment this with a private head, and FedDUET

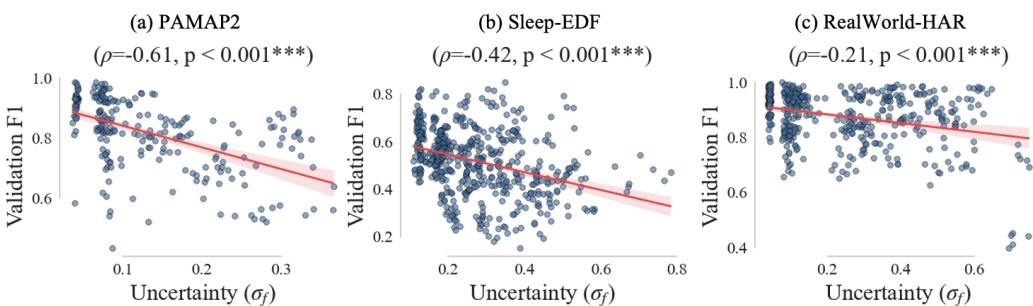

Figure 4: Correlation between multimodal uncertainty ($\sigma_f$) and model performance across three datasets: (a) PAMAP2, (b) Sleep-EDF, and (c) RealWorld-HAR. In all cases, Spearman correlation shows a statistically significant negative relationship, demonstrating that higher predicted uncertainty corresponds to lower F1-scores. This confirms that $\sigma_f$ serves as an effective indicator of client data uncertainty.

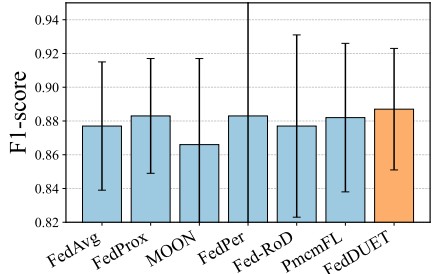

Figure 5: F1-score comparisons with baselines on the Opportunity dataset.

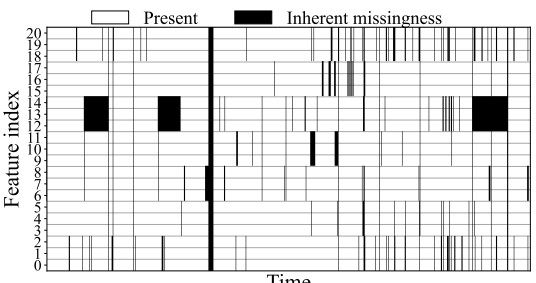

Figure 6: Visualization of inherent missingness patterns in the Opportunity dataset.

further adds lightweight MLP-based uncertainty heads. We train for 200 global rounds, sampling 30~50% of clients for 3 local epochs per round using SGD with momentum 0.9 and weight decay $5 \times 10^{-5}$. Additional experiment details are provided in Appendix E.

## 5.2 RESULTS

**Overall results.** Table 1 presents our main experimental results, evaluating `FedDUET` against baselines under diverse and challenging client heterogeneity and modality instability conditions. The findings demonstrate that `FedDUET` consistently outperforms all competing methods across the three datasets. This is evident in the average F1-scores, where `FedDUET` achieves top performance on PAMAP2 (0.642), Sleep-EDF (0.542), and RealWorld-HAR (0.799), underscoring the broad effectiveness and generalizability of our framework. This consistent superiority stems directly from `FedDUET`'s unique design, which is purpose-built to tackle the dual axes of modality heterogeneity. The framework's resilience to severe intra-client instability (I=Severe) is driven by its Uncertainty-as-Temperature (UT) loss that dynamically modulates predictive entropy, steering predictions toward the true posterior. Simultaneously, its robustness to high inter-client heterogeneity (H=Severe) arises from the Decoupled Training (DT) strategy. By isolating the shared representation from client-specific updates, DT ensures that personalization does not degrade the model's generalizable knowledge—a critical weakness in monolithic approaches. This synergy, where architectural separation provides a stable foundation for fine-grained uncertainty management, is the core reason for `FedDUET`'s consistently superior performance.

**Multimodal uncertainty ($\sigma_f$) as a predictor of performance degradation.** To assess whether the uncertainty predicted by `FedDUET` reliably reflects the uncertainty of client data, we analyze the correlation between multimodal uncertainty ($\sigma_f$) and downstream model performance. We compute the multimodal uncertainty and the corresponding validation F1-score for each client across five seeds and six different missingness settings, and evaluate their Spearman correlation (Spearman, 1961). Figure 4 reports results on three datasets. In all cases, we observe a statistically significant negative

Table 2: Ablation study of `FedDUET`'s core components. All values are reported as macro F1 scores. The top section shows ablated models, while the bottom shows our full model.

| Method Variant | PAMAP2 | Sleep-EDF | RealWorld-HAR |
|---|---|---|---|
| FedDUET w/o UT, DT | $0.608 \pm 0.010$ | $0.520 \pm 0.006$ | $0.765 \pm 0.009$ |
| FedDUET w/o UT | $0.624 \pm 0.011$ | $0.520 \pm 0.008$ | $0.791 \pm 0.008$ |
| FedDUET w/o DT | $0.591 \pm 0.012$ | $0.511 \pm 0.009$ | $0.758 \pm 0.009$ |
| **FedDUET** | $\mathbf{0.642} \pm \mathbf{0.011}$ | $\mathbf{0.542} \pm \mathbf{0.006}$ | $\mathbf{0.799} \pm \mathbf{0.006}$ |

Spearman correlation between $\sigma_f$ and F1-score: $\rho$=-0.61, $\rho$=-0.42, $\rho$=-0.21, and all $p < 0.001$, for PAMAP2, Sleep-EDF, and RealWorld-HAR, accordingly.

This finding confirms that, in general, across all datasets, higher predicted uncertainty is associated with lower predictive performance. The correlation does not simply reflect missingness severity, since removing uninformative signals may not harm accuracy, but instead captures performance-relevant uncertainty. These results demonstrate that `FedDUET` not only adjusts the predictive entropy of the model to better match the true posterior but also produces interpretable uncertainty estimates that closely track downstream reliability. Per-client correlation results for all datasets are provided in Table 6 of Appendix F.

**Evaluation on naturally missing data.** We further evaluate `FedDUET` on the Opportunity dataset (Roggen et al., 2010), which inherently contains missing values rather than simulated dropouts, as illustrated in Figure 6. Figure 5 reports F1-scores across baselines and `FedDUET`. `FedDUET` achieves the best performance with an average F1-score of 0.887 over five seeds. In contrast, Fed-RoD, which ranked second in Table 1 fails to improve over FedAvg, with both yielding an F1-score of 0.877. Under this real-world missingness setting, FedProx and FedPer emerge as the strongest baselines after `FedDUET`, both reaching an F1-score of 0.883.

These results highlight that `FedDUET` consistently achieves the best performance even under real missingness patterns. Importantly, it remains superior despite the Opportunity dataset exhibiting a relatively mild missing rate of about 8%. In addition, `FedDUET` shows lower variance across seeds compared to the baselines, highlighting its robustness and stability in realistic scenarios. Detailed experimental settings are provided in Appendix E.4.

**Ablation study.** Our ablation study in Table 2 dissects the impact of Decoupled Training (DT) and Uncertainty-as-Temperature (UT). Interestingly, we find that introducing the UT loss without a decoupled architecture (`FedDUET` – DT) degrades performance, even falling below the FedAvg (`FedDUET` – UT – DT) baseline. As further evidenced in Appendix F.1, the effectiveness of UT is empirically validated; however, this setting shows that a standard shared model cannot resolve the conflicting uncertainty signals from heterogeneous clients. On the other hand, applying DT alone (`FedDUET` – UT) provides the necessary architectural stability by resolving inter-client heterogeneity, leading to significant gains; however, it does not directly address sample-level intra-client instability. The full `FedDUET` model, which combines both components, achieves the best performance across all datasets. Together, these results validate our design: DT first resolves inter-client heterogeneity, thereby enabling UT to effectively mitigate intra-client instability.

## 6 CONCLUSION

We addressed the dual challenges of intra-client instability and inter-client heterogeneity in multi-modal federated health sensing. We introduced `FedDUET`, a framework that integrates a Decoupled Training (DT) architecture with an Uncertainty-as-Temperature (UT) loss to jointly ensure robust generalization and reliable personalization. Through principled simulation and extensive evaluation across multiple real-world datasets, we demonstrated that `FedDUET` consistently outperforms strong baselines under diverse and realistic missingness regimes. Beyond empirical gains, our findings establish that decoupling shared and private components while explicitly modeling uncertainty are key principles for building the next generation of federated learning systems capable of handling the complexities of multimodal sensing in the wild. We outline our limitations and provide further discussions in Appendix G.

ETHICS STATEMENT

We have used publicly available multimodal health sensing datasets in our experiments. There are no ethical issues with this paper.

REPRODUCIBILITY STATEMENT

We have provided the complete pseudocode of `FedDUET` in Algorithm 1. Experimental and implementation details are included in Appendix E.

USAGE OF LARGE LANGUAGE MODELS

Large Language Models (LLM)s were used in enhancing writing quality of the manuscript through grammar correction and structural sentence reorganization.

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

# FedDUET: Bridging Modality Gaps with Decoupled Uncertainty-Enhanced Training

# Appendix

## A  ALGORITHM

---

**Algorithm 1** `FedDUET`: Decoupled Uncertainty-Enhanced Training

---

1: **Server input:** Initial shared parameters $\theta_G^0$, global rounds $T$, client selection rate $C$.
2: **Client $k$'s input:** Local dataset $\mathcal{D}_k$, local epochs $E$, learning rate $\eta$.

3: **for** $t \leftarrow 0$ **to** $T - 1$ **do**
4:      Sample a client subset $\mathcal{S}_t$.
5:      Communicate $\theta_G^t$ to all clients $k \in \mathcal{S}_t$.
6:      **for** each client $k \in \mathcal{S}_t$ **in parallel do**
7:          // Client-Side Local Training //
8:          Initialize private head $\theta_{P,k}$ if not exists; Set local model $\theta_{G,k} \leftarrow \theta_G^t$.
9:          **for** $e \leftarrow 1$ **to** $E$ **do**
10:              **for** each batch $(x, y) \in \mathcal{D}_k$ **do**
11:                  $z_G, \{z_m, s_m\}, h_f \leftarrow \text{ForwardShared}(\theta_{G,k}, x)$.
12:                  $\sigma_m \leftarrow \exp(s_m/2)$ for each modality $m = 1, \dots, M$.
13:                  $\mathcal{L}_G \leftarrow \text{CE}(z_G, y) + \frac{1}{M}\sum_{m=1}^{M} \mathcal{L}_{\text{UT}}(x_m, \sigma_m, y)$.      ▷ Equation 6
14:                  $\theta_{G,k} \leftarrow \theta_{G,k} - \eta \nabla_{\theta_{G,k}} \mathcal{L}_G$.

15:                  $z_{P,k} \leftarrow \text{ForwardPrivate}(\theta_{P,k}, \text{detach}(h_f))$.
16:                  Fuse $\{\sigma_m\}$ into a multimodal uncertainty $\sigma_f$.
17:                  $\mathcal{L}_{P,k} \leftarrow \mathcal{L}_{\text{mUT}}(x, \sigma_f, y)$.      ▷ Equation 7
18:                  $\theta_{P,k} \leftarrow \theta_{P,k} - \eta \nabla_{\theta_{P,k}} \mathcal{L}_{P,k}$.
19:              **end for**
20:          **end for**
21:          Communicate updated shared parameters $\theta_{G,k}$ to the server.
22:      **end for**
23:      // Server-Side Aggregation //
24:      Partition each $\theta_{G,k}$ into unimodal $\{\theta_{G,k}^{\text{uni},m}\}_{m=1}^{M}$ and multimodal $\theta_{G,k}^{\text{multi}}$ parts.
25:      **for** each modality $m \in \{1, \dots, M\}$ **do**
26:          $\bar{\theta}_G^{\text{uni},m} \leftarrow \sum_{k \in \mathcal{S}_t} w_k^m \theta_{G,k}^{\text{uni},m}$, where $w_k^m \propto |\mathcal{D}_{k,m}|$.
27:          $\theta_G^{\text{uni},m,t+1} \leftarrow (1 - r_m)\theta_G^{\text{uni},m,t} + r_m\bar{\theta}_G^{\text{uni},m}$.
28:      **end for**
29:      $\theta_G^{\text{multi},t+1} \leftarrow \sum_{k \in \mathcal{S}_t} w_k \theta_{G,k}^{\text{multi}}$, where $w_k \propto |\mathcal{D}_k|$.
30: **end for**

31: **Server output:** Final shared parameters $\theta_G^T$.
32: **Client $k$'s output:** Personalized head parameters $\theta_{P,k}$.

---

The complete `FedDUET` framework, which integrates the Uncertainty-as-Temperature loss within our Decoupled Training strategy, is detailed in Algorithm 1. The process unfolds over multiple communication rounds coordinated by a central server. In each round, selected clients receive the current shared model components ($\theta_G$). During local training, each client updates these shared components using the $\mathcal{L}_G$ objective, learning generalized representations that are robust to intra-client modality instability. Concurrently, each client's private head ($\theta_{P,k}$) is updated using the $\mathcal{L}_{P,k}$ objective to specialize for the client's unique modality set, with gradients detached to preserve the integrity of the shared model. Finally, the server aggregates the updated shared parameters from all clients.

To enhance stability, this aggregation is performed in a partitioned manner: unimodal components are updated using a modality-weighted Exponential Moving Average (EMA), while the remaining

multimodal components are aggregated using standard Federated Averaging. Note that a standard EMA smoothing for unimodal encoder parameters is applied uniformly for all baselines for stability. This process produces a robust global model along with specialized private heads tailored to each client's data. During training, we keep the learning objectives for the shared and personal components decoupled; the personal head is trained to directly specialize on the client's data using the shared features, without being influenced by the global model's classification output. At inference time, however, the logits from the generalist global model and the specialist personal head are ensembled via summation, combining their complementary knowledge to produce a more robust and accurate final prediction.

## B  RELATED WORK

**Federated learning in health sensing.**    Federated Learning (FL) (McMahan et al., 2017) offers a compelling solution for data-sensitive domains such as healthcare (Antunes et al., 2022; Dang et al., 2022), enabling training on decentralized data without compromising privacy. FL has been applied to diverse healthcare sensing tasks, including medical image segmentation (Liu et al., 2021), human activity recognition (Ouyang et al., 2021), and mortality prediction (Vaid et al., 2021). However, many of these applications operate under the simplifying assumption that clients possess homogeneous sensor infrastructures and complete modality sets. This assumption is misaligned with real-world deployments, where modality heterogeneity is pervasive due to variations in device ownership and intermittent sensor failures. Importantly, this heterogeneity is not simply another instance of statistical non-IID data, but a structural challenge spanning two distinct axes: intra-client modality instability and inter-client modality heterogeneity. To address this gap, we introduce a principled simulation framework in Section 2 that formalizes and realistically models both challenges.

**Federated learning with multimodal and missing data.**    Work on multimodal FL under missing modalities spans both benchmarks and algorithms, but most evaluations simplify modality missingness and heterogeneity. FedMultimodal (Feng et al., 2023) standardizes tasks and robustness tests, yet models modality availability with a per-modality Bernoulli process at a uniform rate, omitting the temporal burstiness of real sensing streams. Methods designed for non-IID data, such as FedProx (Li et al., 2020) and MOON (Li et al., 2021), improve robustness to distribution shifts but are modality-agnostic and do not address sample-level absence. Personalization approaches based on global–private decoupling (Arivazhagan et al., 2019) or representation decoupling (Chen & Chao, 2022) handle cross-client variation but generally assume complete inputs at each step. Methods tailored to missing modalities, such as (Bao et al., 2023), compensate with priors or surrogates, masking absent representations with learned prototypes to provide global prior information. Reconstruction-based methods (Wang et al., 2024; Zheng et al., 2023) instead synthesize absent inputs or features, but result in huge computation and communication costs. In contrast, our framework explicitly targets *both* axes of heterogeneity by (i) tempering logits with uncertainty to down-weight unreliable, partially observed samples, and (ii) decoupling shared representation learning from client-specific heads guided by uncertainty, thereby avoiding reconstruction and public-data reliance while remaining effective under realistic missingness dynamics.

**Uncertainty estimation in deep learning.**    Uncertainty estimation in deep learning has been extensively studied and is commonly categorized into epistemic and aleatoric uncertainty. *Epistemic uncertainty*, which reflects the model's ignorance about its parameters, is often is addressed at inference time through techniques such as Monte Carlo dropout or deep ensembles (Kendall & Gal, 2017). *Aleatoric uncertainty*, which accounts for inherent noise and ambiguity in the data, is typically modeled by predicting an input-dependent variance alongside the primary output. This variance is then used to down-weight noisy or ambiguous samples, a strategy that has proven effective in regression tasks (Kendall & Gal, 2017) and was later extended to classification for mitigating label noise (Collier et al., 2021; Englesson et al., 2023). In time-series applications such as sensing data, uncertainty modeling has also been applied to imputation for missing values (Kim et al., 2023), but imputers are often inefficient and risk introducing bias. Motivated by the information-theoretic principle that missing data increases posterior entropy ($H(Y|X_{observed}) \geq H(Y|X_{complete})$), we instead use aleatoric uncertainty as an input-dependent temperature to directly calibrate the model's predictive distribution. For intra-client instability, the learned temperature stabilizes local training by modulating gradients for dropout-affected samples. For inter-client heterogeneity, the shared model

learns robust representations together with their associated uncertainty estimates. This uncertainty signal guides the personalization of private heads, enabling them to specialize effectively without corrupting the generalizable shared model. As a result, personalization becomes both modality-aware and reliability-calibrated.

## C  PROOF OF ENTROPY UNDER MISSINGNESS

**Proposition.**  Let the complete data sample $X_c = (X_o, X_m)$, consist of observed $X_o$ and missing $X_m$ parts. Then, in expectation, the entropy of the true posterior with missing inputs is greater than or equal to complete inputs:

$$H(Y \mid X_o) \geq H(Y \mid X_c).$$

**Proof.**  When $X_m$ is missing, the posterior marginalizes over its possible values:

$$p(y \mid X_o) = \int p(y \mid X_o, x_m)\, p(x_m \mid X_o)\, dx_m.$$

This forms a mixture distribution over the complete data posteriors $p(y \mid X_o, x_m)$.

The Shannon entropy $H(\cdot)$ is concave. By Jensen's inequality, the entropy of a mixture distribution is greater than or equal to the expectation of the entropies of its components.

$$H\left(\sum_i \pi_i P_i\right) \geq \sum_i \pi_i H(P_i).$$

Applying this property to the missing data case yields

$$H\big(p(y \mid X_o)\big) \geq \mathbb{E}_{X_m \mid X_o}\big[H\big(p(y \mid X_o, X_m)\big)\big].$$

Finally, taking expectation with respect to $X_o$ gives the conditional entropy inequality

$$H(Y \mid X_o) \geq H(Y \mid X_o, X_m) = H(Y \mid X_c).$$

Therefore, the expected entropy of the posterior under missingness is greater than or equal to that with complete information. □

## D  DETAILS ON MODALITY HETEROGENEITY SIMULATIONS

Table 3: Core properties of the datasets and task setup.

| Dataset | Sampling Rate (Hz) | Window Length |
|---|---|---|
| PAMAP2 | 100 | 200 (2.0s) |
| RealWorld-HAR | 50 | 150 (3.0s) |
| Sleep-EDF | 100 | 3000 (30.0s) |

### D.1  SIMULATION HYPERPARAMETERS

This section details the specific hyperparameter configurations used to generate the simulated datasets for our experiments. The inherent properties of each dataset, including sampling rate and the classification window size, are listed in Table 3.

The simulation parameters are detailed in Table 4. For **Inter-Client Modality Heterogeneity**, the parameters of the Beta($\alpha_a, \beta_a$) distribution are kept consistent across datasets. For **Intra-Client Modality Instability**, we define the expected burst length for operational and missing states, thereby directly modeling realistic sensor failure scenarios.

The underlying Markov chain transition probabilities, $p_{11}$ (present-to-present) and $p_{00}$ (missing-to-missing), are from these expected durations. The probability of remaining in a state is calculated as

Table 4: Hyperparameter configurations for simulating inter-client heterogeneity and intra-client instability.

| Inter-Client Heterogeneity Parameters | | |
|---|---|---|
| **Level** | **Description** | **Beta**$(\alpha_a, \beta_a)$ |
| Homogeneous | All modalities are available. | N/A |
| Moderate | Moderate modality variation. | Beta(45, 20) |
| Severe | High modality variation. | Beta(45, 45) |

| Intra-Client Instability Parameters | | | |
|---|---|---|---|
| **Dataset** | **Level** | **Exp. Operational Burst (seconds)** | **Exp. Missing Burst (seconds)** |
| PAMAP2 | Moderate | 100s | $\sim$33s |
| | Severe | 100s | 100s |
| RealWorldHAR | Moderate | 200s | $\sim$67s |
| | Severe | 200s | 200s |
| Sleep-EDF | Moderate | 1000s ($\sim$17m) | 500s ($\sim$8m) |
| | Severe | 1000s ($\sim$17m) | 1000s ($\sim$17m) |

$p = 1 - (1/L)$, where $L$ is the target expected burst length in time steps ($L$ = duration in seconds $\times$ sampling rate). For example, for PAMAP2, an expected 100-second operational burst corresponds to $L = 100\text{s} \times 100\text{Hz} = 10,000$ steps, yielding a transition probability of $p_{11} = 1 - 1/10,000 = 0.9999$.

### D.2 SIMULATION EXAMPLES ACROSS CLIENTS

Figure 7 illustrates representative examples of simulated missingness patterns across 12 clients on RealWorld-HAR dataset under our proposed framework. Two key properties can be observed.

**Diversity under inter-client heterogeneity.** Even within the same inter-client heterogeneity level (i.e., using identical $(\alpha_a, \beta_a)$ values for the Beta prior), the set of available modalities differs across clients due to the stochastic sampling process. For example, client index 1 exhibits 5 unavailable modalities, whereas client index 4 has only 1 unavailable modality. This variability faithfully reflects realistic deployment scenarios, where individuals may own heterogeneous device configurations with different sensor types and counts.

**Bursty instability under intra-client dynamics.** In addition, the Markov-chain design for intra-client instability introduces temporal burstiness in sensor stability, resulting in partially missing segments of varying lengths across clients and time. Importantly, the simulated patterns qualitatively resemble the real-world missingness observed in the Opportunity dataset (Roggen et al., 2010) (shown Figure 6), where modalities exhibit intermittent, bursty dropouts rather than independent random noise. This alignment highlights that our simulation not only models the static diversity of sensor ownership but also captures realistic temporal instability of sensing streams.

## E EXPERIMENT DETAILS

### E.1 DATASETS

We use the following real-world multimodal health sensing datasets in our experiments: PAMAP2 (Reiss & Stricker, 2012), RealWorld HAR (Sztyler & Stuckenschmidt, 2016), and Sleep-EDF (Goldberger et al., 2000; Kemp et al., 2000).

**PAMAP2** (Reiss & Stricker, 2012) consists of recordings from nine users performing twelve activities using wearable Inertial Measurement Unit (IMU) sensors. Following prior work (Jain et al., 2022), we

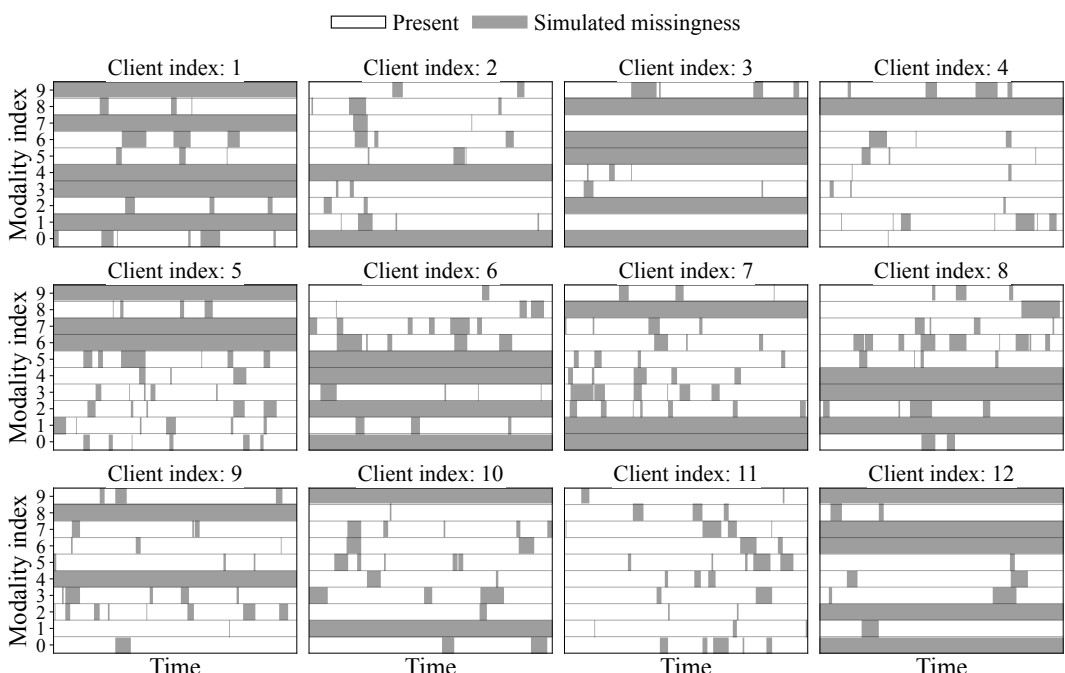

Figure 7: Missing patterns of 12 clients in the RealWorld-HAR dataset under moderate inter-client heterogeneity and intra-client stability.

exclude one subject who contributed data for only a single activity. The dataset provides accelerometer and gyroscope signals from three body locations: wrist, chest, and ankle, yielding six sensing modalities in total.

**Sleep-EDF** (Goldberger et al., 2000; Kemp et al., 2000) contains sleep recordings from 20 participants, including electroencephalography (EEG), electrooculography (EOG), chin electromyography (EMG), Respiration, and event markers. Each recording is annotated with hypnograms containing five sleep stages. Following prior work (Tsinalis et al., 2016; Phan et al., 2018), we utilize the Sleep Cassette subset, which focuses on age-related sleep patterns in healthy individuals.

**RealWorld-HAR** (Sztyler & Stuckenschmidt, 2016) consists of activity recordings from fifteen participants performing eight daily activities. Data were collected with seven body-worn IMU sensors, two of which were discarded due to limited activity coverage. The final dataset comprises signals from ten modalities, spanning five body locations and two IMU sensor types.

### E.2 BASELINES

**FedAvg** (McMahan et al., 2017) represents the foundational approach to FL, enabling decentralized training without sharing raw data. As a baseline framework, FedAvg is crucial for assessing the lowest achievable accuracy, especially in scenarios lacking specific mechanisms to address missing modalities.

**FedProx** (Li et al., 2020) was proposed to address system and statistical heterogeneity. It enhances performance by adding a proximal term to the local training loss, penalizing deviations between local and global models to improve stability and convergence.

**MOON** (Li et al., 2021) targets the problem of local data heterogeneity. It incorporates contrastive learning into federated learning, encouraging alignment between the global and local models' embeddings while pushing apart embeddings from the client's previous local model. MOON has demonstrated strong performance across multiple image classification benchmarks, establishing its effectiveness under non-IID conditions.

**FedPer** (Arivazhagan et al., 2019) addresses statistical heterogeneity by splitting models into shared base layers and client-specific personalization layers. The base layers are trained collaboratively across clients using FedAvg, while the personalization layers are updated only with local data. FedPer improves robustness compared to FedAvg when faced with heterogeneous client distributions.

**FedRoD** (Chen & Chao, 2022) bridges generic and personalized federated learning. It decouples the local model into two predictors: a generic head trained with balanced risk minimization to improve robustness against non-IID class distributions, and a personalized head trained with empirical risk minimization to capture client-specific patterns. Fed-RoD consistently outperforms prior approaches under heterogeneous data conditions.

**PmcmFL** (Bao et al., 2023) introduces a prototype library to address the challenges of missing modalities in federated multimodal learning. Prototypes are used both as masks for absent modalities and as anchors in a contrastive loss to reduce client heterogeneity. This design alleviates task drift and improves robustness, achieving state-of-the-art performance under diverse missing-modality settings.

### E.3 DETAILS OF LEARNING SETUP

Table 5: Hyperparameter configurations for all experiments.

| Method | Hyperparameter | PAMAP2 | RealWorld-HAR | Sleep-EDF |
|--------|----------------|--------|---------------|-----------|
| FedAvg | Learning Rate | 0.001 | 0.03 | 0.03 |
| FedPer | Learning Rate | 0.001 | 0.03 | 0.001 |
| FedProx | Learning Rate 
 Proximal Term ($\mu_{\text{prox}}$) | 0.001 
 0.1 | 0.03 
 0.01 | 0.01 
 0.01 |
| MOON | Learning Rate 
 Contrastive Weight ($\mu_{\text{contrast}}$) 
 Temperature ($\tau$) | 0.001 
 10 
 0.5 | 0.03 
 0.1 
 0.5 | 0.03 
 10 
 1.0 |
| PmcmFL | Learning Rate 
 CLIP Loss Weight | 0.001 
 0.1 | 0.03 
 0.01 | 0.001 
 0.5 |
| FedRoD | Learning Rate | 0.001 | 0.03 | 0.01 |
| `FedDUET` | Learning Rate | 0.001 | 0.001 | 0.001 |

**Details on learning setup.** Table 5 lists the tuned hyperparameters for each method and dataset. For all datasets, we sweep the learning rate over $\{0.001, 0.01, 0.03, 0.05\}$. For FedProx (Li et al., 2020), we tune the proximal coefficient $\mu_{\text{prox}} \in \{0.001, 0.01, 0.1, 0.5, 1\}$. For MOON (Li et al., 2021), we tune the contrastive weight $\mu_{\text{contrast}} \in \{0.1, 1, 5, 10\}$ and the temperature $\tau \in \{0.1, 0.5, 1\}$. For PmcmFL (Bao et al., 2023), we tune the CLIP loss weight over $\{0.01, 0.1, 0.5, 1.0, 5.0\}$.

For model selection, we employ a validation-based approach, which is tailored to the objective of the target method. For standard federated learning methods like FedAvg and FedProx, we adopt a global model selection policy. The server identifies the single global model that achieves the highest average F1-score across all clients' validation sets, and this globally best model is used for the final test evaluation. In contrast, for personalized methods such as FedPer, FedRoD, and our proposed `FedDUET`, we use a local model selection strategy. Each client independently tracks and saves the state of its own personalized model that performs best on its local validation data. Consequently, the final test performance is reported using each client's individually selected best model, aligning the evaluation with the goal of personalization.

### E.4 DETAILS OF EVALUATION ON NATURALLY MISSING DATA.

We use the Opportunity dataset (Roggen et al., 2010), a multivariate time-series dataset collected for human activity recognition using wearable, object, and ambient sensors. It includes five runs per subject of daily activities (ADL runs) in natural settings, alongside a drill run. For our evaluation, we focus on the ADL runs and use four coarse activity labels: Stand, Walk, Sit, and Lie.

From the full suite of sensors, we used seven body-worn inertial measurement units (IMUs): Accelerometer $RKN^\wedge$ (Right Knee, Up), HIP (Hip), $LUA^\wedge$ (Left Upper Arm, Up), $RUA\_$ (Right Lower Arm, Up), LH (Left Hand), BACK (Back), and $RKN\_$ (Right Knee). Each accelerometer provides tri-axial measurements (x, y, z), resulting in 21 feature columns in total. The dataset is partitioned into four clients, with a client selection rate of 100%. The sampling rate is 32 Hz. In total, we obtain 13,537,335 recorded values, among which 1,081,770 entries are missing, corresponding to approximately 8% missingness overall.

# F  ADDITIONAL EXPERIMENT RESULTS

## F.1  EFFECTIVENESS OF THE UNCERTAINTY-AS-TEMPERATURE LOSS

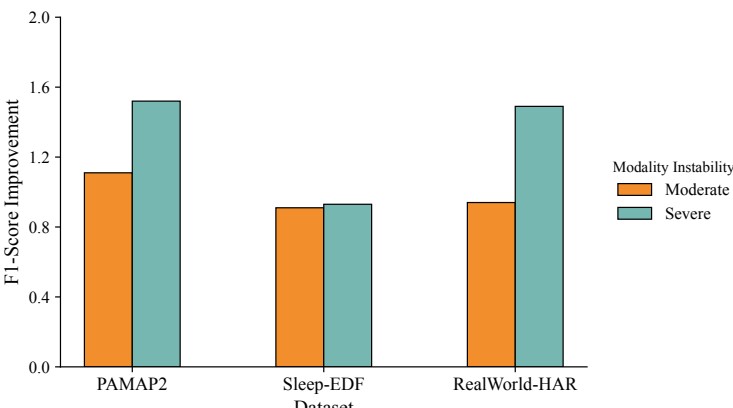

Figure 8: F1-score improvements achieved by replacing cross-entropy loss with the proposed Uncertainty-as-Temperature (UT) loss in a centralized setting. Results are averaged over unimodal experiments on three datasets (PAMAP2, RealWorld-HAR, and Sleep-EDF) under varying levels of modality instability. Performance gains from UT become increasingly pronounced as modality instability worsens.

To assess the effectiveness of our proposed Uncertainty-as-Temperature (UT) loss, we conduct experiments in a centralized setting. Specifically, we replace the standard cross-entropy loss with UT loss and evaluate improvements in F1-score across unimodal settings. Figure 8 reports the average improvements on PAMAP2, Sleep-EDF, and RealWorld-HAR under two levels of modality instability.

Across all datasets, UT consistently improves performance over cross-entropy. The gains become increasingly significant as instability worsens. For example, PAMAP2 and RealWorld-HAR achieve improvements exceeding 1.5% in the severe setting. These results validate our theoretical motivation: UT calibrates predictive distributions by adjusting their entropy with a learned temperature, thereby better matching the true posterior under missing modalities. Its benefits are most pronounced when modality instability is severe.

## F.2  PER-CLIENT CORRELATION ANALYSIS

Table 6 reports the per-client Spearman correlation between multimodal uncertainty ($\sigma_f$) and model performance across the three datasets. Excluding non-significant cases, nearly all clients show statistically significant negative correlations, except c1 in RealWorld-HAR and c12 in Sleep-EDF. This confirms that, for the majority of clients, higher predicted uncertainty reliably corresponds to lower model performance, reinforcing our key finding that uncertainty estimation in `FedDUET` provides an effective measure of client-level data reliability.

Table 6: Per-client correlation results between predicted multimodal uncertainty ($\sigma_f$) and model performance (F1-score) across three datasets: PAMAP2, RealWorld-HAR, and Sleep-EDF. The table reports Spearman correlation coefficients ($\rho$) along with their statistical significance levels.

(a) PAMAP2

| Client | $\rho$ | Significance |
|--------|--------|--------------|
| c1 | -0.565 | $p < 0.01^{***}$ |
| c2 | -0.819 | $p < 0.001^{****}$ |
| c3 | -0.503 | $p < 0.01^{**}$ |
| c4 | -0.326 | n.s. |
| c5 | -0.563 | $p < 0.01^{***}$ |
| c6 | -0.738 | $p < 0.001^{****}$ |
| c7 | -0.477 | $p < 0.01^{**}$ |
| c8 | -0.910 | $p < 0.001^{****}$ |

(b) RealWorld-HAR

| Client | $\rho$ | Significance |
|--------|--------|--------------|
| c1 | +0.601 | $p < 0.001^{***}$ |
| c2 | -0.435 | $p < 0.05^{*}$ |
| c3 | +0.204 | n.s. |
| c4 | -0.607 | $p < 0.001^{***}$ |
| c5 | -0.382 | $p < 0.05^{*}$ |
| c6 | +0.048 | n.s. |
| c7 | -0.634 | $p < 0.001^{***}$ |
| c8 | -0.687 | $p < 0.001^{****}$ |
| c9 | -0.806 | $p < 0.001^{****}$ |
| c10 | -0.954 | $p < 0.001^{****}$ |
| c11 | +0.128 | n.s. |
| c12 | -0.483 | $p < 0.01^{**}$ |
| c13 | -0.366 | $p < 0.05^{*}$ |
| c14 | +0.313 | n.s. |
| c15 | -0.552 | $p < 0.001^{***}$ |

(c) Sleep-EDF

| Client | $\rho$ | Significance |
|--------|--------|--------------|
| c1 | -0.894 | $p < 0.001^{****}$ |
| c2 | -0.821 | $p < 0.001^{****}$ |
| c3 | -0.608 | $p < 0.001^{***}$ |
| c4 | -0.869 | $p < 0.001^{****}$ |
| c5 | -0.078 | n.s. |
| c6 | -0.890 | $p < 0.001^{****}$ |
| c7 | -0.595 | $p < 0.001^{***}$ |
| c8 | -0.287 | n.s. |
| c9 | -0.853 | $p < 0.001^{****}$ |
| c10 | -0.689 | $p < 0.001^{****}$ |
| c11 | -0.452 | $p < 0.05^{*}$ |
| c12 | +0.590 | $p < 0.001^{***}$ |
| c13 | -0.444 | $p < 0.05^{*}$ |
| c14 | -0.336 | n.s. |
| c15 | -0.810 | $p < 0.001^{****}$ |
| c16 | -0.953 | $p < 0.001^{****}$ |
| c17 | -0.959 | $p < 0.001^{****}$ |
| c18 | -0.663 | $p < 0.001^{****}$ |
| c19 | -0.777 | $p < 0.001^{****}$ |
| c20 | -0.796 | $p < 0.001^{****}$ |

## G  LIMITATIONS AND DISCUSSIONS

One consideration for the `FedDUET` framework is the potential system overhead from its additional components, namely the unimodal Uncertainty Heads and the dual G/P-Heads. While these components lead to a slight increase in local computation and communication costs relative to baselines like FedAvg (McMahan et al., 2017), the impact is minimal. The Uncertainty and private P-Heads are intentionally designed as lightweight two hidden layer MLPs, ensuring their computational footprint is negligible. This design choice makes `FedDUET` far more efficient than alternative methods that rely on data imputation or feature reconstruction (Zheng et al., 2023), which are notoriously expensive in both computation and communication. Given the substantial performance improvements from robustly handling heterogeneity, this modest increase in model complexity is a highly effective trade-off.

Furthermore, while our work focuses on federated health sensing, the principles of `FedDUET` are broadly applicable to any domain involving federated learning on multimodal time-series sensing data. For instance, our method could be adapted for tasks such as robust autonomous driving (Prakash et al., 2021), or predictive maintenance in industrial IoT (Zhang et al., 2025). We chose to focus on the healthcare domain because it is an area where the need for privacy-preserving machine learning is paramount. These sensitive nature of health data makes Federated Learning not just a beneficial paradigm but often a necessary one, making it critical application area for developing robust, real-world solutions.

