# OpenReview forum: "FedDUET: Bridging Modality Gaps with Decoupled Uncertainty-Enhanced Training"
_ICLR.cc/2026/Conference — ICLR 2026 Conference Withdrawn Submission_

### Official Review · Reviewer_sVzH · 2025-10-28

**Soundness:** 2
**Presentation:** 2
**Contribution:** 3
**Rating:** 4
**Confidence:** 2

**Summary:**

This paper investigates the problem of federated learning with multimodal health sensing data under realistic modality heterogeneity—both intra-client instability (bursty, temporally correlated sensor dropout within a client) and inter-client heterogeneity (diverse, static sensor suites across clients). The authors introduce a principled simulation framework that systematically captures both axes of real-world modality heterogeneity. The main technical contribution is FedDUET, a method combining Uncertainty-as-Temperature (UT) loss to adapt prediction confidence to data uncertainty and a Decoupled Training (DT) procedure that isolates a shared representation from client-specific heads. Extensive experiments on four real-world datasets demonstrate state-of-the-art results and highlight the value of joint uncertainty modeling and decoupled personalization.

**Strengths:**

1.Rigorous and Realistic Problem Formulation: The paper clearly identifies limitations in prior work's missing data simulation and formalizes a dual-axis model capturing both static (inter-client) and dynamic (intra-client) heterogeneity, as showcased in Figure 1. The Markov and Beta-Bernoulli processes for missing data are principled and well-motivated.
2.Principled Simulation Framework: Figure 2 and Table 4, along with simulation procedures in Sections 2, 4, and Appendix D, enable realistic stress-testing of robustness in FL under missing modalities.
3.Detailed Ablation Analysis: Table 2 establishes the necessity of both UT and DT, as removing either degrades performance (even below baseline in some settings).

**Weaknesses:**

1.Figures Interpretation and Design: Some figures could be clearer/well-annotated (for example, Figure 6’s y-axis is labeled "Feature index", but additional context for the sensor types would increase interpretability).
2.Mathematical Rigor: The entropy arguments in Appendix C, though valid, are somewhat standard and could be cited more concisely (this principle is well-known in information theory and Bayesian treatment of missing data). A more substantial originality would have been a new theoretical result, e.g., proving improved calibration or generalization guarantees under the FedDUET loss, or formalizing convergence behavior under dual-heterogeneity. As it stands, the main mathematical claim (entropy increases with missingness) is descriptive rather than predictive.

3.Lack of Discussion on Communication and System Overhead: While minor increases in computation/communication are mentioned in Appendix G, there is no quantitative assessment. How do the additional heads affect overall system latency, memory, or model size on resource-limited clients? Table 3 and Table 5 provide some info but do not quantify this overhead relative to baselines. In realistic healthcare deployments, overhead can be non-trivial.

**Questions:**

Uncertainty Fusion and Calibration: How sensitive is the multimodal uncertainty score ($\sigma_f$) to miscalibration or noise in the unimodal uncertainty predictions? If certain sensors give poor/overconfident uncertainty estimates, does this undermine the effectiveness of the UT loss? Would an alternative fusion (e.g., learned weights) improve results?

---

### Official Review · Reviewer_hxFC · 2025-10-30

**Soundness:** 2
**Presentation:** 2
**Contribution:** 3
**Rating:** 2
**Confidence:** 4

**Summary:**

This paper tackles the problem of continuous modality missing in signal-based multimodal federated learning, where different clients have different modality subsets, observing dynamic temporal drop-out. The authors introduce a novel approach to simulate missingness along multiple axes including inter- and intra-client instability, tackled by a new client design FedDUET based on uncertainty estimation.

**Strengths:**

* Good simulation setup, reflecting complexity of missing modalities in real-world scenarios.
* Good writing, easy to follow

**Weaknesses:**

* Server aggregation. It is unclear how the framework aggregates clients’ parameters.
* Uncertainty learning. It seems like the output of uncertainty head is the scale parameter for the logit instead of the uncertainty given training losses. There is no learning signal guiding the uncertainty head to capture the uncertainty of each modality.
* Logic of uncertainty estimation seems unconventional: The uncertainty estimated from shared parts are instability of input given the global model, which is aggregated from different clients. This means that if the local samples are different from the global data distribution across clients, the instability increases, making both shared and private parts to focus on these samples. In other words, the learned models would be more biased towards local data and the modality missing patterns[1], making global aggregation such as FedAvg or FedProx significantly harder and degrading the overall performance[1,2].
* Lack of literature review. In terms of missing modality in federated learning, there are some lines of research that the authors do not mention[1,2,3] which explore the complicated missingness simulation as the proposal. Furthermore, since the proposed architecture is somewhat similar to standard approaches in personalized federated learning (as mentioned in the main text - line 298~300), the authors should expand the related works section in this way.
* Lack of baselines. [1] also explores a new way to simulate missingness, how is the proposed approach better. The authors should provide evidence that all prior solutions[1,2,3] fail to handle such new scenarios.
* Lack of experimental results. It is unclear how the simulation parameters (controlling the beta-bernoulli process) are set in all experiments and how these missingness would affect the final performance.

[1] Nguyen et al., Learning Reconfigurable Representations for Multimodal Federated Learning with Missing Data, NeurIPS’25

[2]  Nguyen et al., Fedmac: Tackling partial-modality missing in federated learning with cross modal aggregation and contrastive regularization. NCA’24

[3] Chen et al., FedMSplit: Correlation-adaptive federated multi-task learning across multimodal split networks. ACM SIGKDD’23

**Questions:**

See Weaknesses

---

### Official Review · Reviewer_DuGN · 2025-10-31

**Soundness:** 3
**Presentation:** 2
**Contribution:** 2
**Rating:** 4
**Confidence:** 3

**Summary:**

The research focuses on multimodal federated learning with two types of missingness: (i) intra-client instability (bursty sensor dropouts over time) and (ii) inter-client heterogeneity (different sensor suites among users).

**Strengths:**

Clear articulation of dual-axis heterogeneity; the simulation is more accurate than traditional i.i.d. masking.

UT is easy to implement; DT works seamlessly with normal FL backbones; and stop-gradient prevents client-specific noise from leaking into common parameters.

Multiple datasets (including Opportunity, which has inherent missingness) and severity levels, as well as ablation (UT, DT, or both).

The negative Spearman correlation between 𝜎_𝑓 and F1 supports the argument that it measures difficulty.

**Weaknesses:**

UT+DT feels like a disciplined technical integration rather than a conceptual leap; there are clear connections to uncertainty-weighted CE and temperature calibration.

Ablations lack sensitivity to temperature regularization (e.g., bounding σ), uncertainty fusion (precision-weighted versus alternatives), fusion head capacity, and stop-gradient location.

Absent communication/runtime analysis; additional heads (uncertainty + private) add parameters/compute—how do wall-clock and uplink/downlink scale vs Fed-RoD or PmcmFL?

Claims regarding "better reflecting the posterior" would be stronger with ECE/NLL/Brier measurements rather than merely F1.

Only time-series wearables were studied; it is uncertain if the results apply to vision/text multimodal FL or asynchronous/partial participation FL.

**Questions:**

What is the per-round compute/communication overhead compared to Fed-RoD and PmcmFL (MB transferred, seconds per round, GPU hours)?

How sensitive are precision fusion results to ϵ, uncertainty heads' capacity, and σ cap (e.g., clamping or regularizing s=logσ^2)?

Can you provide ECE/NLL/Brier for the global head, private head, and ensemble (global + private) with/without UT?

What happens if gradients from the private head are allowed to pass to fusion/encoders (partial or full detachment)?

---

### Official Review · Reviewer_g972 · 2025-10-31

**Soundness:** 1
**Presentation:** 2
**Contribution:** 1
**Rating:** 2
**Confidence:** 4

**Summary:**

This paper proposes the FedDUET framework, aiming to address two challenges in multimodal FL: first, intra-client instability, and second, inter-client heterogeneity. The former refers to variations in quality and missingness across different modalities within a single client's data, while the latter concerns different clients possessing different modality combinations, collectively forming a complex cross-client modality heterogeneity challenge. To address this, the paper proposes three strategies: first, in data simulation, using a two-state Markov chain to simulate bursty dropout of modality data; second, proposing Uncertainty-as-Temperature loss to handle the first challenge; third, proposing Decoupled Training to address the second challenge, uploading encoders as shared components to the server to learn general representations, while keeping heads as client-specific private personalization layers. The paper conducts comparisons with 6 FL baselines on 3 health sensing datasets.

**Strengths:**

This paper introduces the research problem of intra-client modality data instability in multimodal FL, which I have not seen in the multimodal FL literature before. However, in centralized multimodal learning, this is already an existing problem (Xu et al., 2025). Besides this, the paper's approach of using a two-state Markov chain to simulate bursty sensor failures appears interesting, as it may more closely approximate real data compared to simple random dropout.

**Weaknesses:**

1. The main claimed contributions of the paper lack novelty and most of what is proposed in this paper already exists. Specifically:
    - The core mechanism of UT loss originates from (Kendall & Gal, 2017), and the paper merely applies it to multimodal scenarios. However, as described in Weakness #2, the paper's implementation has critical differences from the original method, which may lead to training instability.
    - The DT strategy has been extensively studied in FL, such as FedPer (Arivazhagan et al., 2019). Although the paper claims its innovation lies in introducing DT to multimodal scenarios and guiding personalization through uncertainty, almost the same DT method already exist in multimodal FL [1].

2. The mathematical formulation of UT loss has serious flaws and may be fundamentally infeasible for the following reasons:
    - The paper's Equation (3) is $\mathcal{L}_{\text{UT},m} = \text{CE}(z_m/\sigma_m, y)$, missing the regularization term $\log(\sigma^2)$ from (Kendall & Gal, 2017).
    - Without the regularization term, $\sigma_m$ can theoretically approach arbitrary values. When the model makes incorrect predictions, the gradient $\partial \mathcal{L}/\partial \sigma_m$ will drive $\sigma_m \to \infty$ to escape the penalty, causing all logits to be scaled to 0, degenerating the model into uniform prediction.
    - The ablation experiments in Table 2 indirectly support this analysis: FedDUET w/o DT (i.e., using only UT loss) performs even worse than the baseline without UT and DT.
    - The paper claims $\sigma_m$ represents "uncertainty," but there is no evidence showing that the learned $\sigma_m$ truly corresponds to statistical uncertainty rather than simply sample difficulty or arbitrary weight coefficients.

3. The experimental evaluation is insufficient:
    - The paper only compares with one multimodal FL method (PmcmFL). What about others, such as [1-3]?
    - The paper only simulates inter-client heterogeneity and intra-client instability, but these are only modality heterogeneity. What about other types of heterogeneity commonly considered in FL, such as class non-IID, client sample imbalance, client computational/communication heterogeneity, etc.?

[1] Communication-Efficient Multimodal Federated Learning: Joint Modality and Client Selection
[2] Harmony: Heterogeneous Multi-Modal Federated Learning through Disentangled Model Training
[3] Towards Optimal Multi-Modal Federated Learning on Non-IID Data with Hierarchical Gradient Blending

**Questions:**

1. Can you provide plots showing loss convergence and the variation curves of $\sigma_m$? I am wondering about the possibility of $\sigma_m \to \infty$.

2. The paper's proposed two-state Markov chain for modeling the operational state of each sensor may be interesting, and different modeling approaches will certainly affect model training. What is the rationale for setting the Beta parameters here? Will more severe dataset heterogeneity during training lead to better model training? What if the Beta parameters are dynamic, for example, more uniform in the early stages of FL and allowing the model to learn more extreme cases in later stages? What happens if the training and testing distributions are different? Obviously, we cannot assume the distribution of unseen test datasets. In the real world, users have different habits. How should Beta parameters be set to maximize the model's performance generalization?

---

### Note · Authors · 2025-11-14

I have read and agree with the venue's withdrawal policy on behalf of myself and my co-authors.